# Meiosis I progression in spermatogenesis requires a type of testis-specific 20S core proteasome

Qianting Zhang[1], Shu-Yan Ji[2], Kiran Busayavalasa[1], Jingchen Shao [1] & Chao Yu [1]

Spermatogenesis is tightly regulated by ubiquitination and proteasomal degradation, especially during spermiogenesis, in which histones are replaced by protamine. However, the functions of proteasomal activity in meiosis I and II remain elusive. Here, we show that PSMA8-associated proteasomes are essential for the degradation of meiotic proteins and the progression of meiosis I during spermatogenesis. PSMA8 is expressed in spermatocytes from the pachytene stage, and assembles a type of testis-specific core proteasome. Deletion of PSMA8 decreases the abundance of proteasome in testes. Meiotic proteins that are normally degraded at late prophase I, such as RAD51 and RPA1, remain stable in PSMA8-deleted spermatocytes. Moreover, PSMA8-null spermatocytes exhibit delayed M-phase entry and are finally arrested at this stage, resulting in male infertility. However, PSMA8 is neither expressed nor required for female meiotic progression. Thus, meiosis I progression in spermatogenesis, particularly entry into and exit from M-phase, requires the proteasomal activity of PSMA8-associated proteasomes.

[1] Department of Chemistry and Molecular Biology, University of Gothenburg, Gothenburg SE-40530, Sweden. [2] Life Sciences Institute, Zhejiang University, Hangzhou 310058, China. Correspondence and requests for materials should be addressed to C.Y. (email: chao.yu@gu.se)

The degradation of major cellular proteins is catalyzed by proteasomes, through which cells respond to intracellular and extracellular signals to modulate cell cycle progression, growth, and aging[1]. Proteasomes are composed of two functional components, a 20S core particle and a regulatory particle. The 20S core proteasome consists of four stacked heptameric ring structures with an (α1–α7)(β1–β7)(β1–β7)(α1–α7) arrangement, with proteolytic reactions being performed by the inner two catalytic β rings, while the outer two structural α rings interact with the regulatory particles[2,3]. The structures and functions of the 20S core proteasomes are evolutionarily conserved in all eukaryotes. Because of the fundamental roles of 20S core proteasomes in cells, deletion or mutation of the genes encoding these α and β subunits (except for the one encoding subunit α3, *Pre9*) leads to lethality in budding yeast[1,4]. Moreover, the in vivo functions of the 20S core proteasomes in mammals have not been investigated. However, different regulatory particles have been identified: the ubiquitous expressed 19S particle (or PA700) unfolds poly-ubiquitinated proteins into the 20S core; the PA28αβ particle functions in immunocytes to promote antigen presentation; and PA200 is testis specific[5–8]. In mammalian testes in particular, approximately 90% of the proteasomes are capped with PA200 particles[6,9].

Spermatogenesis consists of a series of complicated and highly ordered processes and involves several dramatic morphological changes through spermatogonia differentiation, meiotic recombination in prophase I, cell division in meiosis I and II, and spermiogenesis. Many proteins such as meiotic proteins, core histones, and unnecessary organelles are degraded during spermatogenesis[10,11]. The ubiquitin and proteasome systems are implicated in many aspects of these processes[12]. The most striking event of proteasomal degradation during spermatogenesis is the replacement of core histones by protamine. In round spermatids, the histones are degraded by a type of testis-specific proteasome that is capped with PA200 in an acetylation-dependent manner[6,9]. Ubiquitin modifications of histones by RNF8 and RNF20 are believed to facilitate DNA damage repair and histone removal in testes[13–15]. Moreover, a recent study reported that, in meiotic spermatocytes, proteasomes are recruited to chromosome axes, which is required for the degradation of meiotic proteins to regulate meiotic recombination[16]. However, meiosis I progression in spermatocytes is less affected by PA200 deletion, and the in vivo function of proteasomes in meiotic prophase I is less understood.

Despite the differences in the regulation of M-phase entry between spermatogenesis and oogenesis, the processes that occur in meiotic prophase I, especially regarding the chromosomal behaviors and cellular machineries involved, are similar. In both male and female germ cells, meiosis is initiated by retinoic acid, which induces the expression of a set of meiotic prophase I proteins[17,18]. Following SPO11-induced DNA double-strand break (DSB) generation and MRN complex-mediated end resection, D-loops are formed between homologous chromosomes (homologs) with the assistance of the recombinases RAD51 and DMC1 and the single-stranded DNA-binding RPA-MEIOB-SPATA22 complex[19–23]. These recombinational intermediates are further processed to result in single-end invasions or double Holliday junctions and are resolved as either crossovers or non-crossovers in the presence of ZMM proteins, the MutL complex, and other correlated meiotic proteins[24,25]. These proteins function in a step-by-step manner during the process of meiotic recombination to produce physical links between homologs and to ensure that faithful segregation occurs in meiosis I, but how these prophase I proteins are degraded and the involvement of proteasomes remain elusive.

Mammalian testes express another unique proteasomal α4-like subunit, PSMA8[6,26]. PSMA8, or α4s, is the paralog of PSMA7 (α4) in vertebrates and, as such, shares a high degree of similarity with PSMA7. In this study, we found that PSMA8 is required for male fertility by assembling a type of testis-specific 20S core proteasome, which is essential for the degradation of proteins, such as RAD51 and RPA1. PSMA8 is expressed specifically in spermatocytes from pachytene stage and is required for the assembly/stabilization of proteasomes in testes. PSMA8-deficient spermatocytes exhibit delayed onset of M phase and arrest at M phase, while the processes of meiotic recombination and synapsis are less affected by PSMA8 deletion. Taken together, our data suggest that PSMA8 is substituted for PSMA7 to form a type of testis-specific 20S core proteasome that regulates proper entry into and exit from M phase during spermatogenesis.

## Results

**PSMA8 is expressed in spermatocytes from the pachytene stage.** To explore the functions of proteasomes during spermatogenesis, we analyzed the dynamics of ubiquitination and proteasomes in testes sections. The level of ubiquitinated proteins was high throughout the process of spermatogenesis, especially during the transition from the zygotene stage to the pachytene stage and during the conversion of round spermatids to elongated spermatids (Supplementary Fig. 1a). Accordingly, 20S core proteasomes, detected with an antibody against all α subunits (α sub), were expressed in spermatocytes and spermatids at all stages (Fig. 1a; Supplementary Fig. 1b). The abundance of α sub was increased during testis development from postnatal day 8 (PD8) to PD56 (Fig. 1a). In contrast, with the increase in proteasome abundance during testis development, the level of PSMA7 was not increased (Fig. 1a). In adult testes, PSMA7 was expressed strictly in spermatocytes before the pachytene stage (outer layer of cells in stage I and XI seminiferous tubules), and its expression level was dramatically decreased in pachynema and diplonema spermatocytes, as well as in spermatids (Fig. 1b).

*Psma8* genes are found in many vertebrates (such as mammals, turtles, crocodiles, and some bony fishes) and share high similarity with *Psma7* in terms of their neighboring genes, gene structures, and encoded protein sequences (Supplementary Fig. 2), suggesting that *Psma8* was duplicated from *Psma7* in early vertebrates. Therefore, we postulated that PSMA8 might be substituted for PSMA7 in the formation of core proteasomes in mouse testes. High expression levels of PSMA8 in testes or, more specifically, in spermatocytes and spermatids, were detected by reverse transcription polymerase chain reaction (RT-PCR) and western blotting (Fig. 1a; Supplementary 3a, b). PSMA8 was first expressed in spermatocytes at the pachytene stage (weakly in early pachynema and strongly in late pachynema), and its expression persisted thereafter throughout spermatogenesis (Fig. 1a, c). The antibodies used were capable of distinguishing PSMA7 and PSMA8 from each other (Supplementary Fig. 2c). The expression of PSMA8 was consistent with the increased level of α sub in PD21 testes (Fig. 1a), suggesting that, in mammalian testes, there is a type of testis-specific 20S core proteasome in which PSMA7 is replaced by PSMA8. This phenomenon is not unique to mammals, which express two isoforms of the α4 subunit, PSMA7 and PSMA8. In several Drosophila species, including *Drosophila melanogaster*, different isoforms of the proteasomal α4 subunits are specifically expressed in testes (Supplementary Fig. 2c)[27,28]. On the other hand, genes encoding α4 subunits are selectively eliminated in some vertebrates; for example, *Psma8* has been eliminated in amphibians, lizards, birds, and some bony fishes, and *Psma7* has been lost in some other bony fishes (Supplementary Fig. 2d).

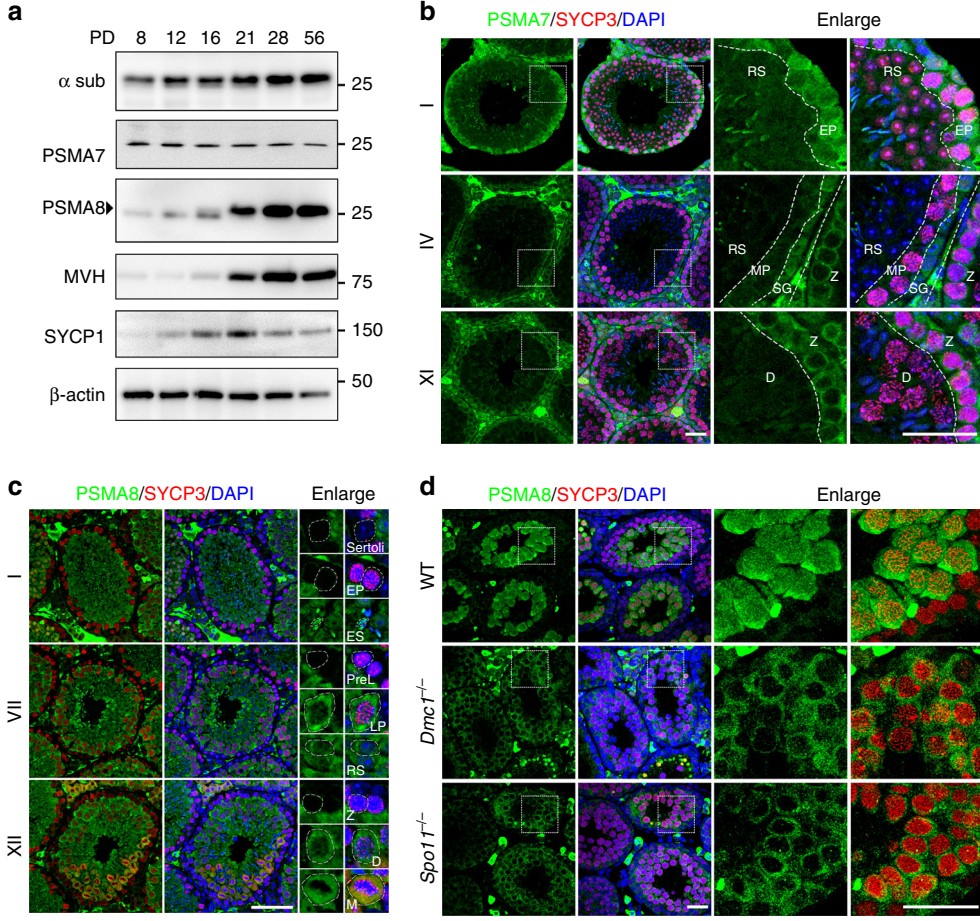

**Fig. 1** PSMA8 is expressed in spermatocytes from the pachytene stage. **a** Western blotting results showing the expression of PSMA7 (α4) and PSMA8 (α4s) during spermatogenesis. The anti-α sub antibody detects all α subunits of the 20S core proteasomes. MVH is a germ cell marker. The arrowhead indicates the specific band. The molecular weights (kDa) are indicated on the right. **b** Immunofluorescent staining of PSMA7 in sections of PD42 wild-type (WT) testes. The regions within the squares are enlarged on the right, and different types of cells were separated by dashed lines. Different stages of seminiferous tubules are shown. RS round spermatids, EP early pachynema, MP mid-pachynema, SG spermatogonia, Z zygonema, D diplonema. Three testes were analyzed. Scale bars, 25 μm. **c** Co-staining of PSMA8 (green) and SYCP3 (red) in sections of WT testes. The representative cell types are enlarged on the right and indicated by dashed circles. Sertoli Sertoli cells, ES elongated spermatid, PreL pre-leptonema, LP late pachynema, M M phase. Scale bar, 50 μm. **d** Immunofluorescent staining of PSMA8 (green) and SYCP3 (red) in sections from WT, Spo11$^{-/-}$, and Dmc1$^{-/-}$ testes at PD21. "−" represents the knockout allele and therefore "−/−" means knockout. Scale bars, 25 μm

Notably, the expression of PSMA8 is consistent with the major repair of DSBs and achievement of full synapsis at the pachytene stage. In Dmc1$^{-/-}$ testes ("−" represents the knockout allele, while "+" represents the wild-type (WT) allele), spermatocytes fail to repair DSBs because of defects in meiotic recombination and are arrested at a zygotene-like stage[29]. Similarly, Spo11$^{-/-}$ spermatocytes are arrested at a pachytene-like stage due to insufficient DSB generation[30]. Both Dmc1$^{-/-}$ spermatocytes and Spo11$^{-/-}$ spermatocytes are arrested before the pachytene stage in stage IV seminiferous tubules. In these Dmc1$^{-/-}$ or Spo11$^{-/-}$ spermatocytes, PSMA8 was expressed at a dramatically lower level (Fig. 1d; Supplementary Fig. 3d).

**PSMA8 is required for male fertility.** To elucidate the direct role of PSMA8-associated 20S core proteasomes during spermatogenesis in vivo, we utilized the CRISPR/Cas9 method to generate mice that were null for Psma8 (Fig. 2a). Following PCR amplification and sequencing of the founder mice, one allele harboring a 5-bp deletion in exon 2 was identified (Fig. 2a; Supplementary Fig. 4a). The founder mouse was further backcrossed to WT mice for three generations to avoid off-target effects. Following heterozygous to heterozygous crossing, Psma8$^{+/-}$ (heterozygous)

and Psma8$^{-/-}$ mice were obtained according to the expected Mendelian ratios and showed no obvious phenotypes with respect to viability and development (Supplementary Fig. 4b–d).

However, Psma8$^{-/-}$ males were infertile (Supplementary Fig. 4c). Western blotting and immunofluorescent staining confirmed that the expression of PSMA8 was completely abolished in the testes of Psma8$^{-/-}$ males (Fig. 2b, c; Supplementary Fig. 4e). We sacrificed juvenile and adult males at different ages and examined their testis size. As shown in Fig. 2d, e, Psma8$^{-/-}$ mice exhibited dramatically smaller testes than WT controls from PD21. The adult Psma8$^{-/-}$ testes (PD90) were approximately 60% smaller than those of the WT controls (Fig. 2e). This difference was apparent as early as PD21, the time point at which the first wave of spermatocytes produce spermatids in WT testes. These results demonstrate that the expression of PSMA8 in spermatocytes is essential for male fertility.

**PSMA8 is required for 20S core proteasome assembly.** Because PSMA8 is the predominant α4 isoform that assembles the 20S core proteasome in testes, deletion of PSMA8 led to an increased level of ubiquitinated proteins, as detected by a ubiquitin (Ub)

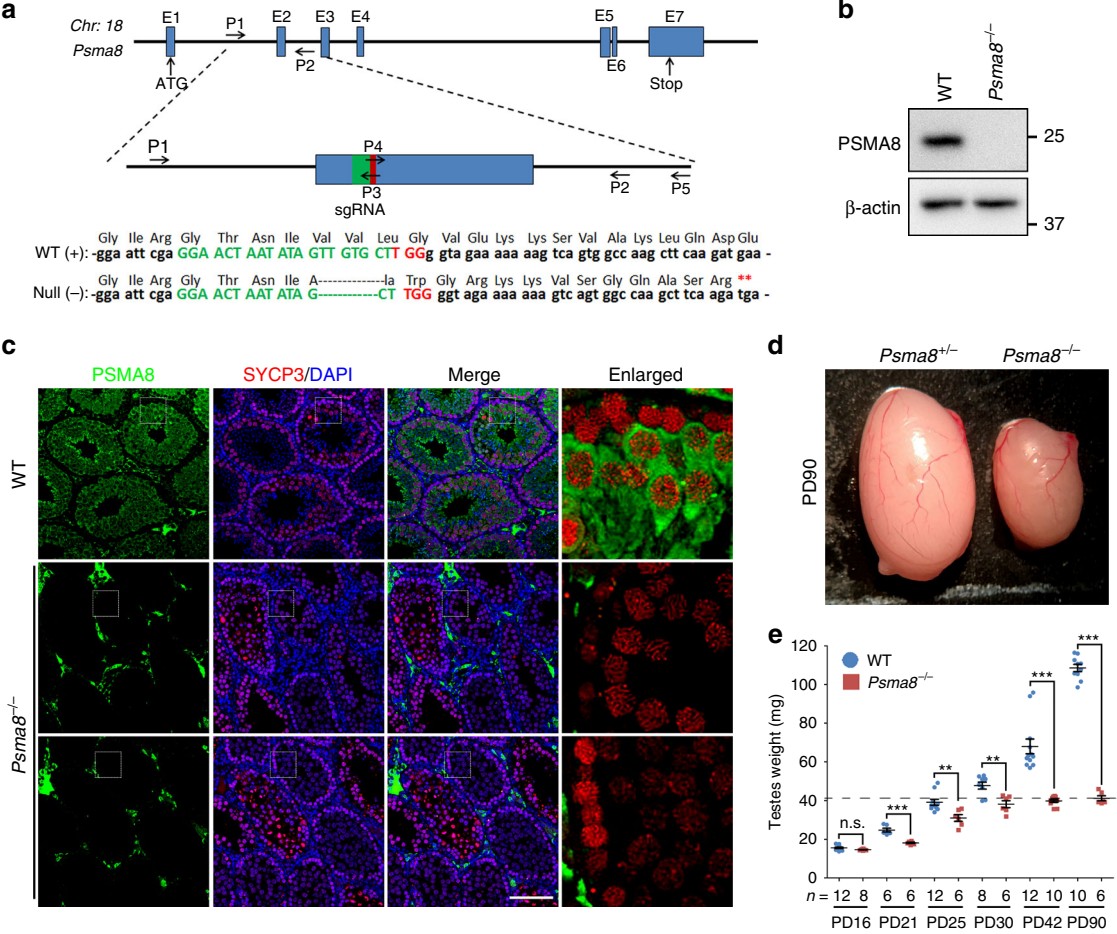

**Fig. 2** PSMA8 is required for male fertility. **a** Schematic diagram showing the gene structure of *Psma8* and the CRISPR/Cas9 strategy used to generate the knockout allele. The null allele of *Psma8* harbored a 5-bp deletion within the selected single-guide RNA (sgRNA) and introduced a premature stop codon (\*\*). The locations of sgRNA and primers (P1–P5) are indicated. The primer sequences are provided in Supplementary Table 1. **b**, **c** Western blotting (**b**) and immunofluorescent staining (**c**) showing successful deletion of PSMA8 in spermatocytes at PD42. Scale bar, 100 μm. **d** A representative image showing the morphology of testes derived from *Psma8*+/− and *Psma8*−/− males at the age of PD90. "+" represents the wild-type (WT) allele. **e** Weights of testes derived from WT and *Psma8*−/− males at the indicated ages. *n* = 6 testes for both WT and *Psma8*−/− at PD21, and *Psma8*−/− at PD25, PD30 and PD90; *n* = 8 testes for WT at PD30 and *Psma8*−/− at PD16; *n* = 10 testes for WT at PD90 and *Psma8*−/− at PD42; *n* = 12 testes for WT at PD16, PD25, and PD42. Error bars indicate S.E.M. \*\**P* < 0.01 and \*\*\**P* < 0.001 by two-tailed Student's *t* tests. n.s. not significant. The dashed line shows the weight of knockout testes at the age of PD90

antibody at PD21 (Fig. 3a), suggesting that proteasomal activity is decreased upon PSMA8 deletion. Interestingly, when stained with the antibody against proteasomal α subunits, the level of 20S core proteasomes was found to be significantly decreased in PSMA8-null pachynema spermatocytes (Fig. 3b), suggesting that PSMA8 is required for the assembly of proteasomes in the testis. The decrease in proteasomal α sub was also confirmed by western blotting of testis samples at the age of PD42 (Fig. 3c). Therefore, we infer that PSMA8 assembles a type of testis-specific 20S core proteasome in pachynema spermatocytes.

On the other hand, the protein level of PSMA7, or α4, was not dramatically affected by PSMA8 deletion in the testes (Fig. 3c). In both WT and *Psma8*−/− testes, PSMA7 was decreased in pachynema spermatocytes (Fig. 3d). Interestingly, in *Psma8*−/− testes, the distribution pattern of α sub (Fig. 3b) was similar to the expression pattern of PSMA7 (Fig. 3d). This correlation implies that the remaining proteasomes in PSMA8-null testes contain PSMA7, thus corresponding to the classic 20S core proteasome. Because the level of PSMA7 was high in pre-leptonema spermatocytes but decreased in pachynema spermatocytes (Fig. 3c, d), we propose that the abundance of these classic 20S core

proteasomes was decreased at the pachytene stage. Therefore, we infer that, at the pachytene stage, the PSMA8-associated 20S core proteasomes replace PSMA7-associated 20S core proteasomes.

**Increased protein stability in PSMA8-null spermatocytes.** RAD51 functions in homologous recombination in both mitosis and meiosis by stabilizing resected single-stranded DNA and facilitating the formation of D-loops[31,32]. As shown in Fig. 4A, RAD51 was expressed at a relatively high level in spermatocytes at the zygotene and pachytene stages. In these cells, RAD51 is required to repair the DSBs generated at the leptotene stage and is localized to the sites of recombination in the nuclei of these meiocytes (Fig. 4a). However, RAD51 was found to be degraded in diplonema spermatocytes (diplotene in stage XI tubules and diakinesis in stage XII tubules), before entering M phase of meiosis I (Fig. 4a). Replication protein A (RPA1/2/3) binds and stabilizes single-stranded DNA during DSB repair in both mitosis and meiosis. Thus RPA1 was highly expressed and localized to the recombinational intermediates in spermatocytes at both the zygotene and pachytene stages (Fig. 4b). Similar to RAD51, RPA1

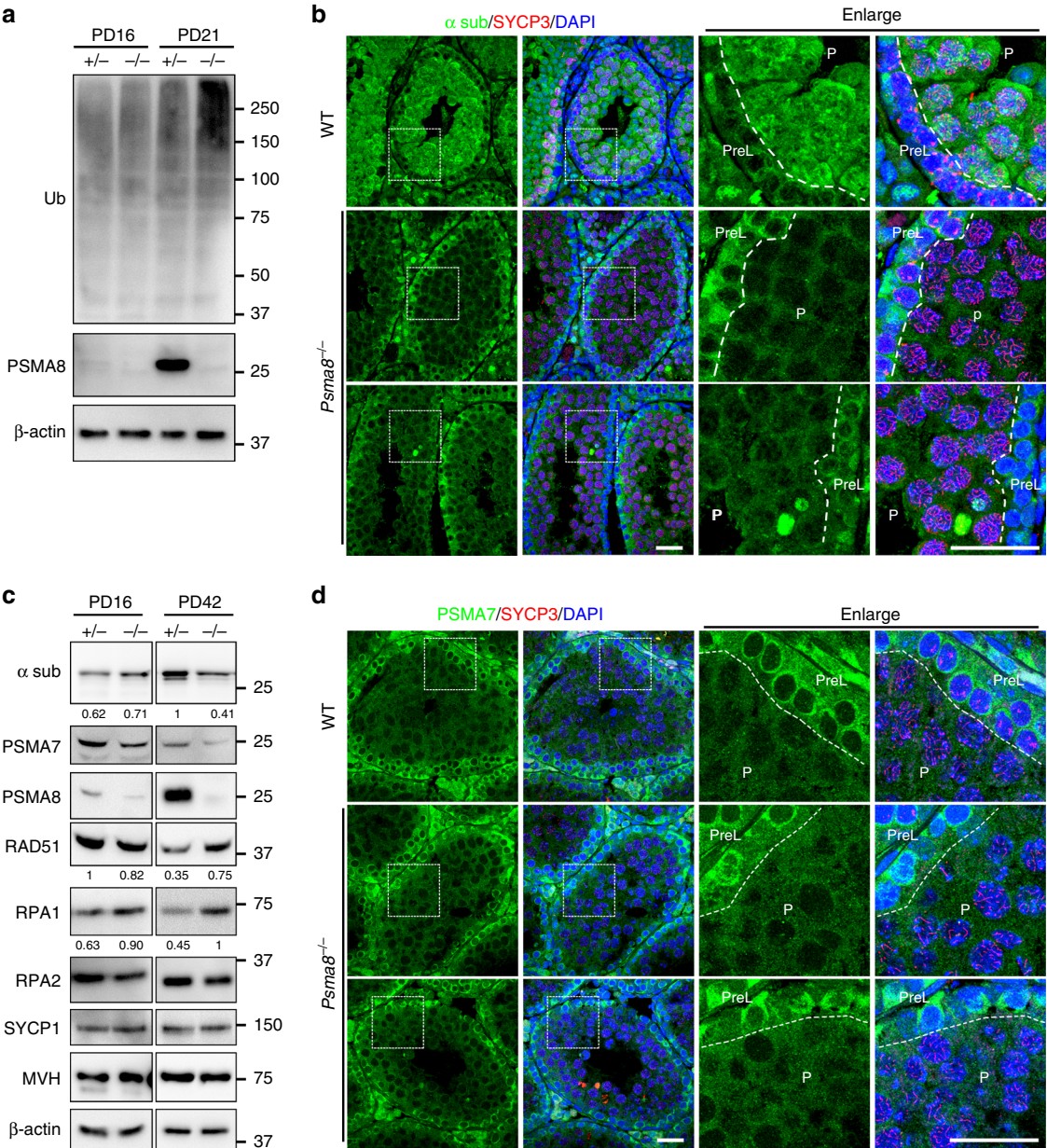

**Fig. 3** PSMA8 is required for proteasome assembly and protein degradation. **a** Western blotting results showing the elevated ubiquitination level in *Psma8*−/− testes at the age of PD21. **b** PSMA8 deletion destabilized the proteasomes, as shown by α sub immunofluorescent staining in sections of PD25 testes. P pachynema. Scale bars, 25 μm. **c** Western blotting results showing the instability of proteasomes and failure of their ability to degrade prophase I proteins (RAD51 and RPA1) in *Psma8*−/− testes at the age of PD42. The band intensity from three independent western blottings are quantified and the relative values are shown underneath each band. **d** Immunofluorescent staining of PSMA7 in testes sections derived from WT and *Psma8*−/− males at PD42. Scale bars, 25 μm

was degraded in spermatocytes from the diplotene stage (Fig. 4b). These results indicate that certain prophase I proteins, such as RAD51 and RPA1, are efficiently degraded at the end of prophase I during spermatogenesis.

However, some of these prophase I proteins, such as RAD51 and RPA1, remained stable in *Psma8*−/− testes (Fig. 3c). When testes sections derived from WT and *Psma8*−/− males were stained for RAD51 and RPA1, both proteins were found to be retained in *Psma8*−/− spermatocytes at the diplotene stage, whereas in WT spermatocytes, RAD51 and RPA1 were efficiently degraded at the same stage (Fig. 4c, d). The elevated level of RAD51 was due to defects in proteasomal degradation in PSMA8-null spermatocytes, rather than changes in mRNA levels (Supplementary Fig. 5a, b). Next, we overexpressed PSMA7 and

PSMA8 in *Psma8*−/− testes. PSMA8 overexpression resulted in the degradation of RAD51 in *Psma8*−/− spermatocytes, whereas overexpression of PSMA7 did not (Fig. 4e). Similarly, overexpressed PSMA8 decreased the level of RAD51 in WT testes (Fig. 4f; Supplementary Fig. 3c).

Some other prophase I proteins, such as SPATA22, which forms a complex with RPA proteins, and two ZMM proteins, TEX11 and MZIP2, which promote crossover formation[33–35], were degraded normally in spermatocytes before the diplotene stage in a PSMA8-independent manner (Supplementary Fig. 5c–e), suggesting that the degradation of SPATA22 and TEX11 is independent of PSMA8-associated 20S core proteasomes. Thus we infer that PSMA7-associated classic proteasomes are also functional in spermatogenesis.

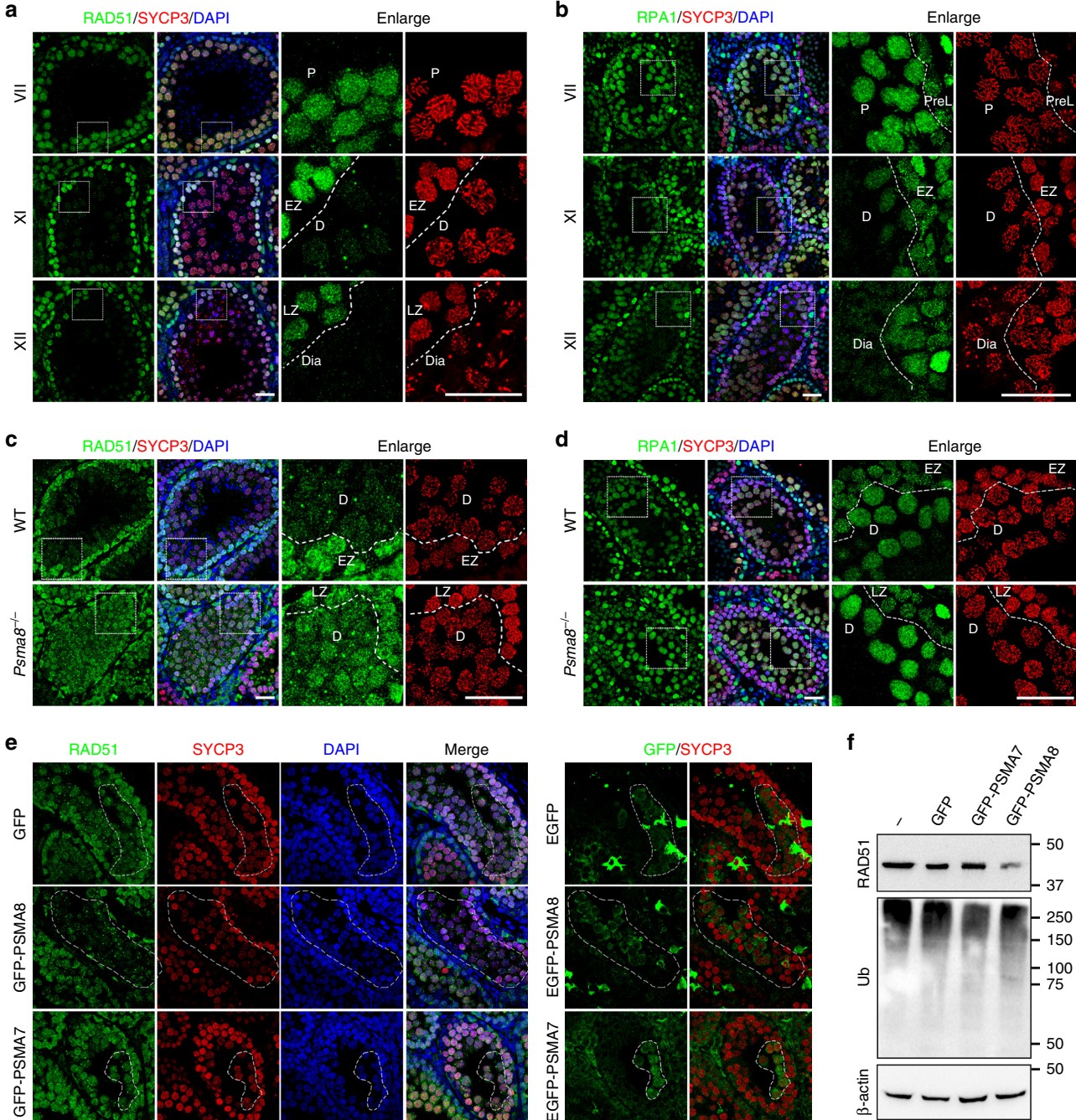

**Fig. 4** Failure in RAD51 and RPA1 degradation in PSMA8-deleted spermatocytes. **a, b** Immunostaining of RAD51 (**a**) and RPA1 (**b**) in PD25 testes sections. SYCP3 (red) and DAPI (blue) staining showing the stages of spermatogenesis. The different stages of seminiferous tubules are indicated by roman numerals. The regions bordered with a dashed box are enlarged on the right two panels. Scale bars, 25 μm. EZ early zygonema, Dia diakinesis spermatocytes. **c–d** Immunostaining of RAD51 (**c**) and RPA1 (**d**) in testes sections derived from wild-type and *Psma8$^{-/-}$* males at the age of PD25. Scale bars, 25 μm. **e** Immunostaining of RAD51 in sections of testes electroporated with plasmids encoding green fluorescent protein (GFP), GFP-PSMA8, and GFP-PSMA7, respectively. Immunostaining of GFP was shown on the right to indicate the GFP-expressing cells, which were bordered with dashed circles. Scale bars, 50 μm. **f** Western blotting showing the levels of RAD51 and ubiquitination in testes electroporated with GFP, GFP-PSMA7, or GFP-PSMA8

**PSMA8-null spermatocytes fail to progress through meiosis I.** However, the processes of DSB generation and repair (γH2AX; Fig. 5a), synapsis (SYCP1 and SYCP3; Fig. 5b), meiotic recombination (RAD51; Fig. 5c, d), and crossover formation as indicated by the crossover-specific late recombination marker, MLH1 (Fig. 5e, f), in meiotic prophase I were less affected by the deletion of PSMA8. PSMA8-null spermatocytes progressed to a late-pachytene stage similar to WT spermatocytes (Supplementary Fig. 6a). Moreover,

the size of adult *Psma8$^{-/-}$* testes (39.8 ± 0.87 mg; $n = 10$; "±" represents stand error of the mean or S.E.M) was larger than the testes of males derived from synapsis- and recombination-defective knockout models (23.8 ± 0.39 mg for *Spo11$^{-/-}$* and 23.5 ± 0.73 mg for *Dmc1$^{-/-}$*; $n = 10$; PD42), in which spermatogenesis was arrested at the pachytene stage. We infer that the major defects in PSMA8-deleted spermatocytes might occur later than the pachytene stage. Hematoxylin & eosin (H&E)

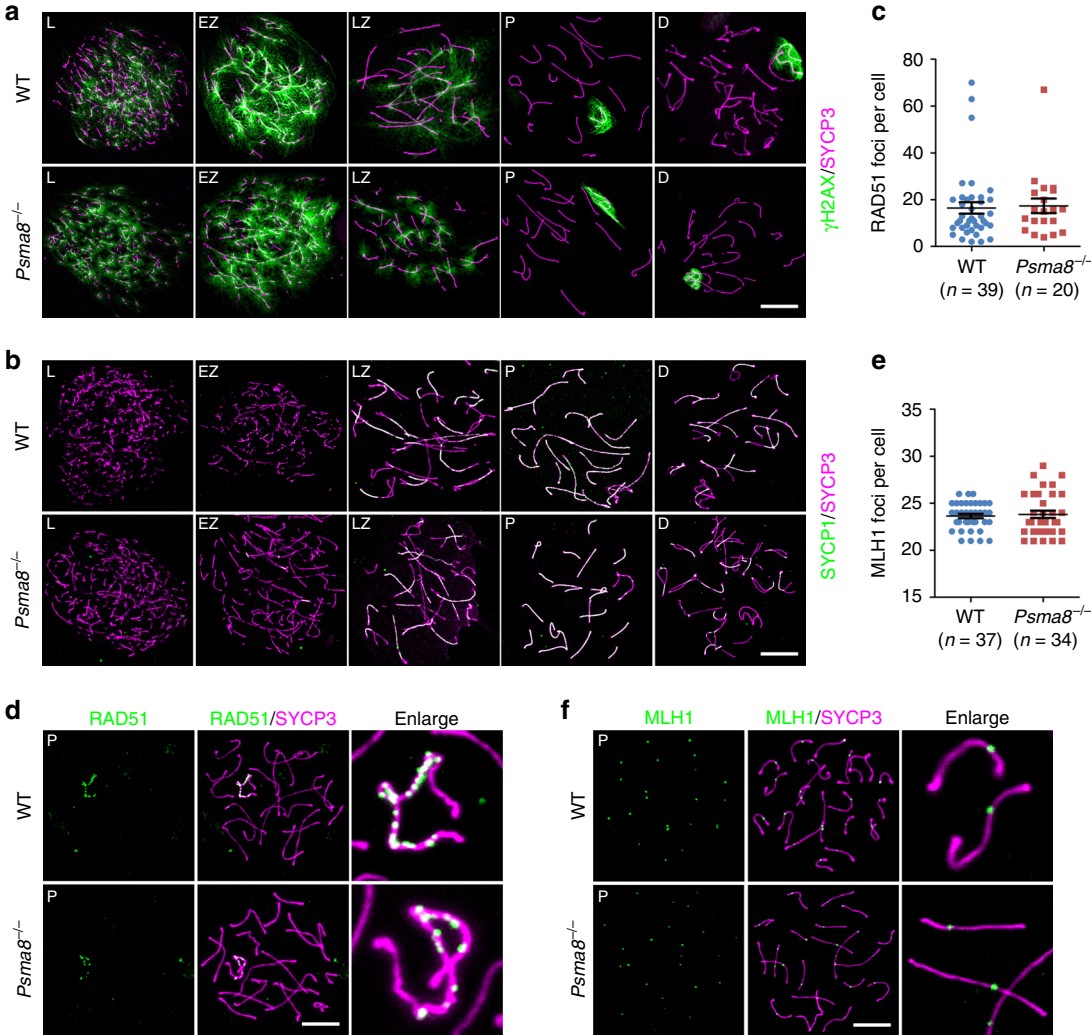

**Fig. 5** Meiotic recombination is less affected by PSMA8 deletion. **a**, **b** Staining of γH2AX (**a**) and SYCP1 (**b**) in nuclear surface spreads derived from wild-type (WT) and $Psma8^{-/-}$ males at PD42. Scale bars, 10 μm. **c**, **d** Immunostaining of RAD51 (**d**) in the nuclear surface spreads derived from WT and $Psma8^{-/-}$ males at PD42 and the quantification of RAD51 foci (**c**) in WT and $Psma8^{-/-}$ spermatocytes at pachytene stage. Scale bar, 10 μm. $n = 39$ for WT spermatocytes and $n = 20$ for $Psma8^{-/-}$ spermatocytes. Median focus numbers are marked. Error bars indicate S.E.M. **e**, **f** Immunostaining of MLH1 (**f**) in nuclear surface spreads derived from WT and $Psma8^{-/-}$ males at PD42. The quantification of MLH1 foci in WT and $Psma8^{-/-}$ spermatocytes at the pachytene stage is shown in **e**. Scale bar, 10 μm. $n = 37$ for WT spermatocytes and $n = 34$ for $Psma8^{-/-}$ spermatocytes. Median focus numbers are marked. Error bars indicate S.E.M.

staining of testes sections derived from WT and $Psma8^{-/-}$ males at PD42 showed that PSMA8-deleted spermatocytes were arrested at M phase (Fig. 6a; Supplementary Fig. 6b). Neither spermatids in seminiferous tubules nor sperm in the epididymis were observed in $Psma8^{-/-}$ males (Fig. 6a; Supplementary Fig. 6b).

Histone H3 is phosphorylated at serine 10 (pHH3) during M phase both in mitosis and meiosis. In PD21 WT testes, approximately 12% of the seminiferous tubules contained spermatocytes (but not the mitotic spermatogonia in the outer layer) that were positive for pHH3 (Fig. 6b, c). Considering the fact that no spermatocytes were positive for pHH3 at PD16, spermatocytes actively progressed from prophase I to M phase between the ages of PD16 and PD21 (Fig. 6b). At later stages (PD30 and PD42), spermatocytes in different seminiferous tubules repeatedly entered and exited M phase, resulting in fluctuation in the number of pHH3-positive tubules from 6% to 10% (Fig. 6b). However, almost no tubules were found to contain pHH3-positive spermatocytes in $Psma8^{-/-}$ testes at PD21 (Fig. 6b, c). Thereafter, $Psma8^{-/-}$ spermatocytes gradually

entered M phase, suggesting that the onset of M phase is delayed in these spermatocytes (Fig. 6a, b).

Moreover, in $Psma8^{-/-}$ testes, diplonema spermatocytes together with early-pachynema spermatocytes or pachynema spermatocytes were frequently observed in the same seminiferous tubules (Fig. 6d). This is unusual, because in WT testes, early-pachynema spermatocytes are only found together with round spermatids in stage I seminiferous tubules, and diplotene spermatocytes are only found in stage XI and stage XII seminiferous tubules together with late-zygonema spermatocytes (Fig. 6d)[36]. Therefore, we propose that progression to M phase is delayed in spermatocytes that are null for PSMA8. However, PSMA8-deleted spermatocytes cannot progress further, as spermatids were not observed in $Psma8^{-/-}$ testes (Fig. 6). As a consequence, these M-phase-arrested spermatocytes undergo apoptosis (Supplementary Fig. 6c, d). Taken together, these finding indicate that spermatocytes lacking PSMA8-associated 20S core proteasomes exhibit delayed progression to M phase and are arrested at this stage, resulting in male infertility in these mice. Interestingly, round spermatids, which were never observed

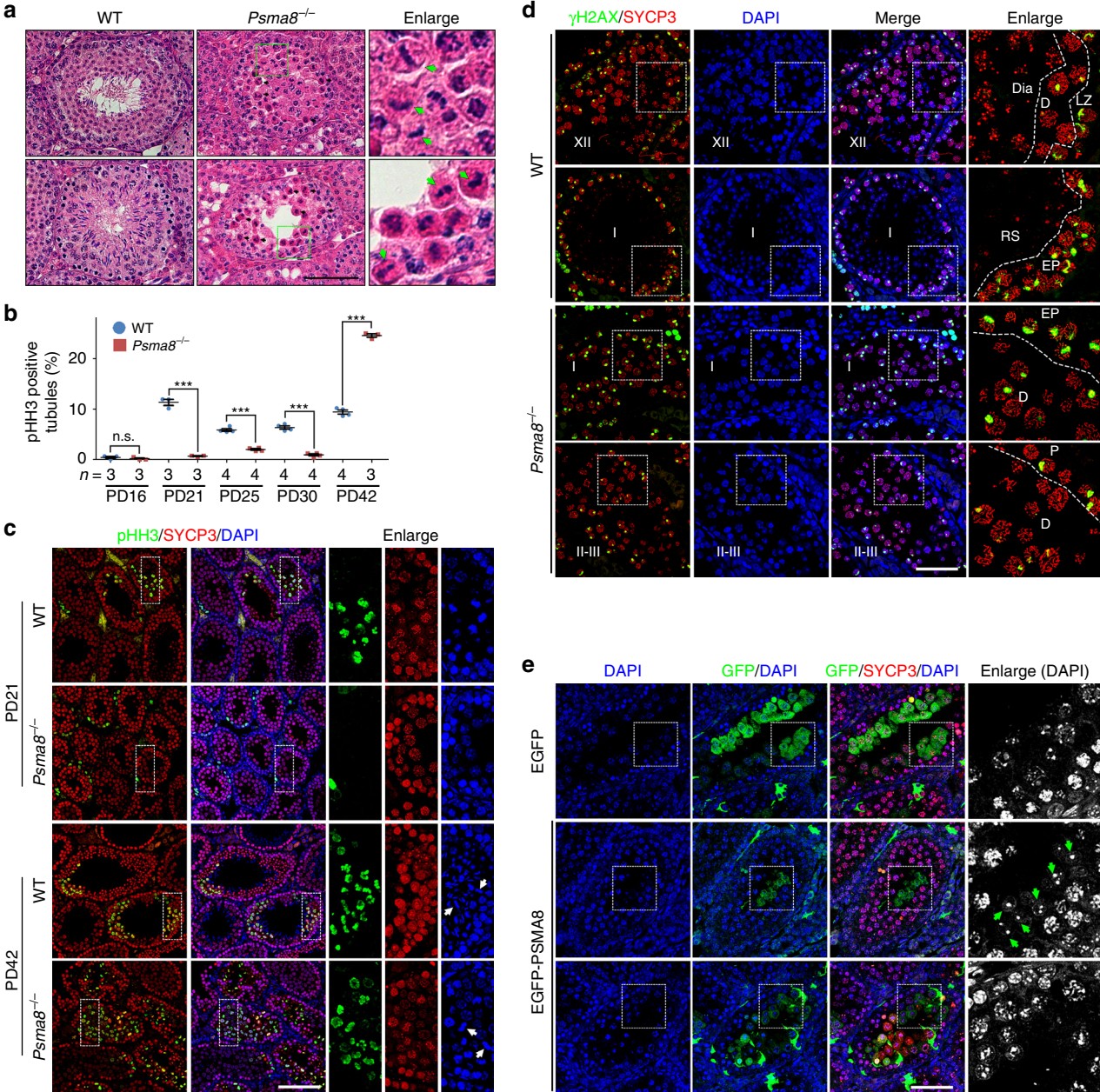

**Fig. 6** PSMA8 deletion causes delayed M-phase entry and M-phase arrest. **a** Hematoxylin & eosin (H&E) staining of testes derived from wild-type (WT) and *Psma8⁻/⁻* males. Some metaphase cells are enlarged on the right and indicated with arrows. Scale bar, 50 μm. **b** Percentage of seminiferous tubules containing spermatocytes positive for phosphorylated Histone H3 (pHH3). *n* = 3 sections for both WT and *Psma8⁻/⁻* testes at PD16 and PD21, and *Psma8⁻/⁻* testes at PD42; *n* = 4 sections for both WT and *Psma8⁻/⁻* testes at PD25 and PD30, and WT testes at PD42. Error bars indicate S.E.M. \*\**P* < 0.01 and \*\*\**P* < 0.001 by two-tailed Student's *t* tests. n.s. not significant. **c** Staining of pHH3 on testes sections derived from WT and *Psma8⁻/⁻* males at PD21 and PD42. Arrows indicate metaphase cells. Scale bar, 100 μm. **d** Staining of phosphorylated H2AX (γH2AX) in testes sections derived from WT and *Psma8⁻/⁻* males at PD42. The stages of seminiferous tubules are indicated. Scale bar, 50 μm. **e** Overexpression of PSMA8 partially rescued spermatogenesis defects. Green arrows indicate round spermatids. Scale bar, 50 μm

in PD21 *Psma8⁻/⁻* testes, could be found in *Psma8⁻/⁻* testes overexpressing PSMA8 (Fig. 6e).

The activity of maturation-promoting factor (MPF complex), which consists of cyclin-dependent kinase 1 (CDK1) and cyclin B, is crucial for prophase–anaphase progression at M phase of both meiosis and mitosis. During the prophase to metaphase transition, CDK1 is activated by phosphorylation on threonine 161 (pT161-CDK1). In *Psma8⁻/⁻* testes, CDK1 is normally activated in spermatocytes at pachytene/diplotene stages and M

phase (Supplementary Fig. 6e). Western blotting results showed that CDK1 was hyperactivated in PSMA8-null testes (Supplementary Fig. 5e). Phosphorylation on threonine 14 and tyrosine 15 represses CDK1 activity, while accumulation of cyclin B1 is required for MPF activation. The levels of pY15-CDK1 and cyclin B1 were not affected by PSMA8 deletion (Supplementary Fig. 5e). EMI2 was recently reported to regulate the prophase I to M-phase transition during spermatogenesis[37]. However, the levels of EMI2 were comparable in WT and *Psma8⁻/⁻* testes (Supplementary

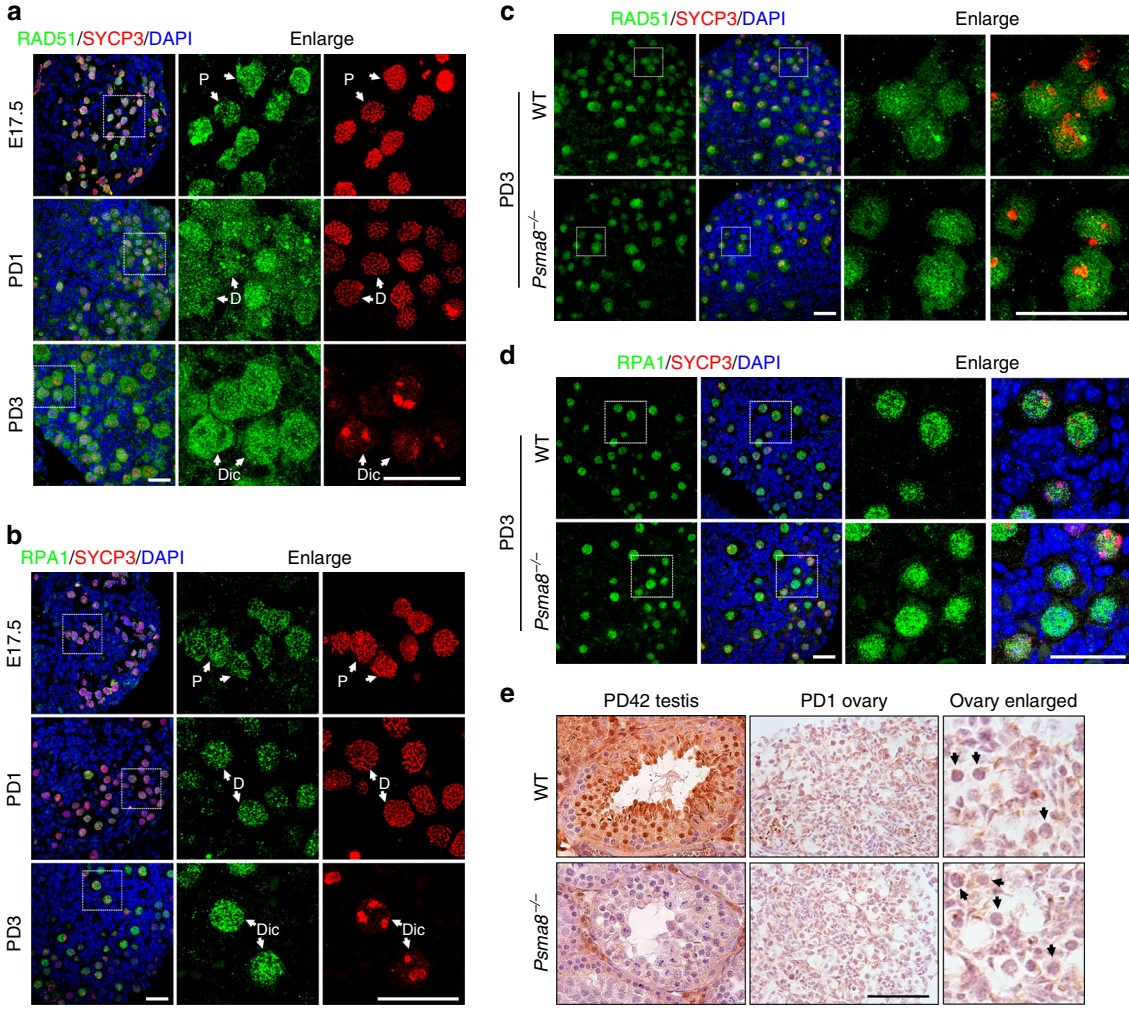

**Fig. 7** PSMA8 is not required for female fertility. **a**, **b** Immunostaining of RAD51 (**a**) and RPA1 (**b**) in ovaries at the indicated ages. Scale bars, 25 μm. Oocytes at different stages are indicated with white arrows. Dic dictyate. **c**, **d** Immunostaining of RAD51 (**c**) and RPA1 (**d**) in wild-type (WT) and *Psma8*$^{-/-}$ ovary sections at PD3. Scale bars, 25 μm. **e** Immunohistochemistry (IHC) staining of PSMA8 in ovary and testes sections derived from WT and *Psma8*$^{-/-}$ mice. Black arrows indicate oocytes. Black arrows indicate individual oocytes. Scale bar, 50 μm

Fig. 5e). Taken together, these results suggest higher MPF activity in PD42 *Psma8*$^{-/-}$ testes, which is consistent with the M-phase arrest phenotype (Fig. 6).

**Female fertility does not require PSMA8**. Intriguingly, PSMA8 is dispensable for female fertility (Supplementary Fig. 4c). In sharp contrast to the finding in males that RAD51 and RPA1 were degraded at the diplotene stage, RAD51 and RPA1 remained stable in diplonema and dictyate oocytes in PD1 and PD3 ovaries (Fig. 7a, b). The protein levels of RAD51 and RPA1 were comparable in dictyate-arrested oocytes in WT or *Psma8*$^{-/-}$ ovaries (Fig. 7a–d; Supplementary Fig. 7). Immunohistochemistry (IHC) showed that PSMA8 was not expressed in oocytes (Fig. 7e). Therefore, we conclude that PSMA8 assembles a type of testis-specific proteasome that can actively degrade RAD51 and RPA1 at the end of meiotic prophase I during spermatogenesis. The activity of PSMA8-associated proteasomes is crucial for efficient prophase I to M-phase progression in male germ cells.

## Discussion

Here we report that, while meiotic proteins such as RAD51 and RPA1 are degraded in diplonema spermatocytes, the levels of these proteins are stable during oogenesis and remain high in the oocytes of primordial follicles. This difference is caused by the assembly of the testis-specific 20S core proteasome, in which PSMA7 is replaced by PSMA8. PSMA8-associated proteasomes are assembled in spermatocytes from the pachytene stage and are capable of degrading meiotic proteins to promote meiosis I progression during spermatogenesis. However, PSMA8 is not expressed in meiotic oocytes, and consequently, the levels of RAD51 and RPA1 are stable. Female fertility is not affected by PSMA8 deletion, suggesting that the progression of meiosis I in oocytes is differentially regulated.

Although both PA200 and PSMA8 are predominantly expressed in testes and assemble the core elements of their respective proteasomes, their functions are unique to spermatogenesis. While PSMA8-associated 20S core proteasomes are crucial for the degradation of prophase I proteins, as we observed, and their deletion causes meiotic arrest at M phase of meiosis I, PA200-capped proteasomes are responsible for the acetylation-dependent turnover of histones during spermiogenesis and DSB repair[6]. *Pa200*$^{-/-}$ males are infertile because they produce malformed spermatids or sperm. However, PA200-null spermatocytes develop normally through meiotic prophase I, meiosis I, and meiosis II and produce haploid spermatids. Therefore, the defects observed in *Psma8*$^{-/-}$ spermatocytes (a longer diplotene stage and M-phase

 9

arrest) probably do not occur in *Pa200*[−/−] spermatocytes. Thus we infer that the PA200 cap is not essential for the activity of the testis-specific 20S core proteasomes that are capable of degrading pro-phase I proteins to promote meiosis I progression.

Because PSMA8-null spermatocytes are arrested at M phase and fail to produce spermatids, investigation of the roles of PSMA8-associated proteasomes in histone turnover is not possible. However, PSMA8 is commonly found in all types of proteasomes in testes, including 20S core proteasomes, 26S proteasomes, and PA200-capped proteasomes[6,26]. We infer that PSMA8 may also participate in this process. Thus a conditional knockout model would be helpful to study the functions of PSMA8 in histone turnover.

On the other hand, we have also observed a subset of proteins whose degradation is independent of PSMA8-associated 20S core proteasomes, such as TEX11, SPATA22, and MZIP2. Removal of these proteins occurs similarly in WT and PSMA8-null spermatocytes (Supplementary Fig. 5c–e). Moreover, the degradation of PSMA7 in pachynema spermatocytes also seems to be independent of PSMA8 (Fig. 3d). Notably, proteasomes and PSMA7 were detected in PSMA8-null testes, although at a decreased level. We infer that the classic proteasomes containing PSMA7 might be functional in these PSMA8-deleted spermatocytes. Proteins such as TEX11, SPATA22, and PSMA7 are degraded by this kind of proteasome. However, the underlying rules regarding substrate selection by PSMA7- or PSMA8-associated proteasomes require further characterization.

Therefore, according to previous studies and our study, we infer that there are at least three kinds of proteasomal degradation events in testes. First, a group of proteins (TEX11, SPATA22, PSMA7, etc.) functioning in meiotic recombination are degraded by the classic 26S proteasomes around the pachytene stage, independent of PSMA8. Second, RAD51 and RPA1 are degraded by PSMA8-associated proteasomes at late prophase I. The third kind of degradation event is that the core histones are replaced by protamine in an acetylation-dependent manner, which is mediated by PA200-capped proteasomes.

## Methods

**Mice**. Mice carrying the *Psma8*-null allele were generated in this study by using CRISPR/Cas9 technology according to standard protocols[38,39]. The diagram of the gene structure and targeting strategy are provided in Fig. 3a. The guide RNA sequence (5′-GGAACTAATATAGTTGTGTGCT-3′) was predicted with an online tool and was cloned into pUC57 by *Bsa*I digestion. The plasmids encoding *sgRNA* and *Cas9* mRNA were linearized and transcribed in vitro, followed by purification according to the manufacturer's instructions (Ambion, AM1345, AM 1354, AM1908 and QIAGEN, 74104). Mouse zygotes were obtained by superovulation of 7–8-week-old C57BL/6 females, which were mated to males of the same strain after hCG administration. *Cas9* mRNA (40 ng/µl) and *sgRNA* (40 ng/µl) were mixed and injected into zygotes with the Eppendorf TransferMan NK2 system. After injection and 2 h of recovery in KSOM medium, zygotes were transferred to pseudopregnant ICR female mice (15–25 zygotes per mouse, both sides). The founder mice were subjected to PCR amplification and sequencing and were crossed with WT mice for three generations to eliminate possible off-target effects. Mice containing null alleles for *Dmc1* and *Spo11* were purchased from Jackson Laboratory and were described previously[30,40].

All mice were maintained under specific pathogen-free conditions in a controlled environment at 20–22 °C, with a 12/12-h light and dark cycle, 50–70% humidity, and food and water provided ad libitum. All animal experimental procedures were approved by the Gothenburg Regional Animal Ethics Committee and were performed in accordance with the guidelines of the University of Gothenburg, Sweden. All mutant mouse strains had a C57BL/6 background. The genotyping primers used are listed in Supplementary Table 1.

**Semi-quantitative RT-PCR**. The embryonic ovaries were collected from embryonic females at embryonic day 16.5. The other tissues were harvested from adult male mice. Male germ cells were prepared by using a bovine serum albumin (BSA) gradient[41]. Total RNA was extracted using the RNeasy Mini Kit (Qiagen, #74106) according to the manufacturer's instructions and reverse-transcribed to obtain first-strand cDNA (Bio-Rad, # 1708890). PCR was performed with Taq DNA polymerase under standard conditions for 25–30 cycles. *Gapdh* served as a loading control. The sequences of the primers used are listed in Supplementary Table 1.

**Histological analyses and IHC staining**. To achieve better histology of testis seminiferous tubules, testes were fixed in Bouin's solution and subjected to H&E staining. For IHC staining, samples were fixed in 3.7% phosphate-buffered saline (PBS) buffered formalin. After fixation, the tissues were dehydrated through an ethanol gradient and embedded in paraffin. The samples were sectioned at a thickness of 5 µm. For H&E staining or IHC, sections were deparaffinized and rehydrated. For IHC, sections were treated with 3% hydrogen peroxide for 10 min at room temperature and with 20 mM sodium citrate for 15 min at 95 °C, followed by overnight cooling. Primary antibodies were applied at suitable dilutions (Supplementary Data 1) at room temperature for 1 h, followed by incubation with biotinylated secondary antibodies for 30 min. Sections were then stained using Vectastain ABC and DAB peroxidase substrate kits (Vector Laboratories, Burlingame, CA). The antibodies used in this study are listed in Supplementary Data 1.

**Cloning and electroporation**. cDNAs encoding *Psma8* and *Psma7* were PCR amplified from mouse testis cDNA and cloned into a pCAG-GFP vector (Addgene #11150) for C-terminal green fluorescent protein (GFP) fusion. After the maxi-prep procedure (Qiagen, #12163), 50 µg of plasmids were injected into live mouse testes at PD16–18. One hour after injection, the testes were electroporated via four forward and four reverse pulses (50 with 950 ms intervals) at a voltage of 30 V. Mice were sacrificed 7 days after electroporation and subjected to preparation of western blotting samples and testes sections.

**Nuclear surface spreading**. Seminiferous tubules were prepared from juvenile and adult testes. Nuclear surface spreads were prepared as described elsewhere[42]. In brief, seminiferous tubules or embryonic ovaries were treated with hypotonic buffer (30 mM Tris, 50 mM sucrose, 17 mM trisodium citrate dehydrate, 5 mM ethyle-nediaminetetraacetic acid, and 0.5 mM dithiothreitol, pH 8.2) for 30 min and subsequently smashed in 100 mM sucrose buffer (pH 8.2). The suspension was then added to slides containing fixative buffer (1% paraformaldehyde and 0.15% Triton X-100, pH 9.2). After at least 2 h of incubation in a humidified box, the slides were air dried and washed.

**Immunofluorescent staining and imaging**. For nuclear surface spreads, slides were blocked with 1% BSA in PBST (PBS with 0.1% Tween-20) for 30 min and subsequently subjected to incubation with primary and secondary antibodies. For paraffin-embedded sections, sections were treated with 20 mM sodium citrate at 95 °C, followed by overnight cooling. The primary antibodies used are listed in Supplementary Data 1, and secondary antibodies with minimal cross-reactivity were purchased from Jackson Immunoresearch Laboratories. The signals were examined under a DeltaVision microscope, and images for quantification were obtained with this microscope. Representative images were obtained by using a confocal laser scanning microscope (Zeiss LSM 700, Carl Zeiss AG, Germany).

**Western blotting**. Testes were lysed directly in 2-mercaptoethanol containing sodium dodecyl sulfate (SDS) loading buffer and heated at 95 °C for 5 min. SDS-polyacrylamide gel electrophoresis and immunoblotting were performed following standard procedures using a semi-dry transfer system (Bio-Rad). The antibodies used are listed in Supplementary Data 1. Uncropped images of western blotting results are shown in Supplementary Fig. 8.

**Statistics**. The experiments were reproduced at least twice with similar results. The results were provided as the means ± S.E.M. Results for two experimental groups were compared by two-tailed unpaired Student's *t* tests. Statistically significant values of $P < 0.05$, $P < 0.01$, and $P < 0.001$ are indicated by *, **, and ***, respectively.

**Reporting summary**. Further information on research design is available in the Nature Research Reporting Summary linked to this article.

## Data availability
The authors declare that all data supporting the findings of this study are available within the article and its supplementary information files or from the corresponding author upon reasonable request.

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

## Acknowledgements
We acknowledge Dr. Heng-Yu Fan, Dr. Marc Pilon, and Dr. Peter Carlsson for discussions. We thank Mary Ann Handel for the H1T antibody. We sincerely acknowledge Dr. Kui Liu for support in the form of funds (VR: 621-2014-5830 and 521-2012-2841) and equipment use at the early stage of this study and for discussions. We acknowledge the Centre for Cellular Imaging at the University of Gothenburg and the National Microscopy Infrastructure, NMI (VR-RFI 2016-00968) for providing assistance in microscopy.

## Author contributions
C.Y. designed the experiments. C.Y. and Q.Z. performed the experiments and analyzed the data. S.-Y.J. performed the microinjection to generate the knockout allele. K.B. generated the mouse antibody against PSMA8 and constructed the plasmids encoding sgRNA. J.S. performed the isolation of male germ cells. C.Y. and Q.Z. wrote the manuscript with help from all authors.

## Additional information

**Competing interests:** The authors declare no competing interests.

