## [Peer Review File · Nature Communications]

Reviewers' comments:

Reviewer #1 (Remarks to the Author):

This reviewer is not yet convinced of the difference in substrate specificity between PSMA7 and PSMA8; more specifically, whether RAD51 and RPA1 is a specific substrate of PSMA8. In figure 4E, what this reviewer like to see is an image of GFP-PSMA8/7 and whether GFP-PSMA8-expressing cells have smaller amounts of RAD51 compared with non-expressing cells and whether expression of PSMA7 affects the mount of RAD51. This point is essential. In addition, to confirm the decrease in RAD51 protein is mediated by proteasomal degradation, the authors should examine whether the mRNA level of RAD51 is not affected and whether treatment with a proteasome inhibitor abolishes the decrease of RAD51 protein. Also, in Figure 4F, immunoblots for GFP and ubiquitin are needed to evaluate the expression level of GFP-PSMA8/7 and to see whether PSMA8/7 expressions assemble functional proteasomes and decrease global ubiquitinated proteins.

Reviewer #2 (Remarks to the Author):

The revision is very responsive and has adequately addressed my concerns through revision and additional data. The data interpretation was appropriate. Over-interpretation has been corrected. The conclusions are appropriate.

Minor textual corrections are needed. For example,
Line 98: change "in conflict with" to "In contrast with"
Line 159: "corelation" to "correlation"
Line 203: "arrested be" to "arrested at"
etc.

Reviewer #3 (Remarks to the Author):

The main change in the revised submission is a focus on male gametogenesis and the removal of unsupported extrapolations to female meiosis. As before, the authors report that the testis-specific proteasomal subunit, PMSA8, is expressed from late zygotene and replaces PMSA7 in assembling the proteasome from late-pachytene stage. Following CRISPR-mediated knockout of PMSA8, Rad51 and Rpa1 become stabilised beyond the time that they would normally disappear in control spermatocytes.

Comments:

1. The authors conclude that "PSMA8 assembles a type of testis-specific proteasome, which can actively degrade RAD51 and RPA1 at the end of meiotic prophase I during spermatogenesis. This is a crucial step for the efficient prophase I to metaphase progression in male germ cells." (line 253).

This conclusion therefore has two components, firstly, that PMSA8 is testis-specific proteasome that degrades Rad51 and Rpa1 and, secondly, that stabilised Rad51 and Rpa1 compromise efficient prophase I-to-metaphase I progression. Their data support the first component. However, as with the previous submission, it is very unconvincing that persisting stability of these two proteins compromises progression. Indeed, as they find in oocytes, Rad51 and Rpa1 ordinarily remain stable in normally progressing oocytes (Fig. 7).

2. The authors describe male infertility following loss of a testis-specific proteasome. However, by focusing on males, the broader interest issue of how males and females differentially regulate gametogenesis is lost.

3. Although the authors show that spermatogenesis is blocked and leads to male sterility when PSMA8 is lost, it is not at all clear why this comes about. The explanation involving Cdk1 remains very tenuous. Whilst the authors have included blots for additional Cdk1 regulators in the revised

version (Fig. S4c), these data do not at all clarify how the supposed phenotype of delayed M-phase entry followed by M-phase arrest comes about or how this might be linked to deregulated proteolysis. Why is entry into M-phase delayed if Cdk1 undergoes normal activation based on pHH3 staining?

4. The authors make conclusions regarding expression levels of proteins and show selected images of immunostains and Westerns. Rather, mean levels from multiple blots etc. should be quantified and depicted graphically, this would greatly simplify the interpretation. For instance, the authors claim that cyclin B1 levels don't change when PSMA8 is deleted; however, the band intensity of the -/- sample in PD21 seems more intense than for the +/- sample.

5. Although the focus is now on males, the authors suggest in the Discussion that "Because spermatocytes null for PSMA8 are delayed in progressing into metaphase I, we postulate that high levels of RAD51 and RPA1 in oocytes might contribute the dictyate arrest (or GV arrest) during the processes of follicle formation." This is unfounded as spermatogenesis in testes is in no way comparable to the G2-arrest in oocytes that involves a unique inhibitory follicular environment for each individual oocyte combined with oocyte-specific APC-Cdh1-mediated cyclin B proteolysis.

Overall, the authors show that knocking out PSMA8 results in a late spermatogenic block and male infertility. This is based on good-quality immunofluorescence but at this stage does not provide any clear insight into why proteasomal disruption in this particular model leads to a meiotic arrest in M-phase.

Reviewer #1 (Remarks to the Author):

This reviewer is not yet convinced of the difference in substrate specificity between PSMA7 and PSMA8; more specifically, whether RAD51 and RPA1 is a specific substrate of PSMA8. In figure 4E, what this reviewer like to see is an image of GFP-PSMA8/7 and whether GFP-PSMA8-expressing cells have smaller amounts of RAD51 compared with non-expressing cells and whether expression of PSMA7 affects the mount of RAD51. This point is essential.

Response:

- 1) In both WT and PSMA8-KO testes, some of meiotic proteins, such as TEX11, SPATA22 and MZIP2 were degraded during testes development (Fig. S4e), suggesting that these proteins could be degraded by the remaining PSMA7-associated proteasomes in PSMA8-null testes. However, since RAD51, RPA1 and CFP1 remained stable in PSMA8-null testes at PD42, we propose that PSMA7-associated proteasomes are unable to degrade them. Instead, the degradation of these proteins at late prophase I is mediated by PSMA8-associated proteasomes.
- 2) As to the RAD51 degradation, because both the GFP antibody and the RAD51 antibody immunofluorescence staining are produced in rabbit, we are not able to co-stain RAD51 and GFP on the same sections. Instead, we stained RAD51 and GFP on the neighboring sections to show the protein level of RAD51 and the expression of GFP-tagged proteins.
- 3) In the revised Fig. 4e, we showed RAD51 and GFP staining side by side. GFP staining on the right indicated the cells expressing GFP-tagged proteins, which were bordered by dashed lines. Therefore, for the left panels showing the protein levels of RAD51, cells within the dashed lines were considered as GFP-expressing cells. As shown in Fig. 4e, we found that cells expressing PSMA8 had lower level of RAD51 than the cells expressing GFP or GFP-tagged PSMA7, suggesting the specificity of PSMA8-associated proteasomes in testes.
- 4) Moreover, we have also overexpressed PSMA7 and PSMA8 in WT testes. As shown in Fig. 4f, overexpression of PSMA7 and PSMA8 both reduced the overall level of ubiquitinated proteins. However, expression of only PSMA8 decreased the protein level of RAD51 in WT testes, suggesting that RAD51 is degraded more efficiently by the PSMA8-associated proteasomes.

In addition, to confirm the decrease in RAD51 protein is mediated by proteasomal degradation, the authors should examine whether the mRNA level of RAD51 is not affected and whether treatment with a proteasome inhibitor abolishes the decrease of RAD51 protein. Also, in Figure 4F, immunoblots for GFP and ubiquitin are needed to evaluate the expression level of GFP-PSMA8/7 and to see whether PSMA8/7 expressions assemble functional proteasomes and decrease global ubiquitinated proteins.

Response:

- 1) We have examined the mRNA level in WT and KO testes during revision. As shown in the revised Fig. S4a, the mRNA level of *Rad51* is not significantly affected by PSMA8-

- deletion, suggesting that the higher RAD51 level in *PsmA8* KO testes is not due to change in mRNA level.
- 2) As the reviewer suggested, we have treated WT and PSMA8-null spermatocytes with MG132, the result of which is shown in the revised Fig. S4b. In WT testes, proteasome inhibition increased the protein level of RAD51, as well as the overall ubiquitination level (Ub), suggesting that the degradation of RAD51 in the end of prophase I requires proteasomal activity.
 - 3) In the revised Fig. 4f, we added the Western blot results for overall ubiquitination. The expression of GFP-tagged proteins was presented in Fig. S2d.

Reviewer #2 (Remarks to the Author):

Minor textual corrections are needed. For example,
Line 98: change "in conflict with" to "In contrast with"
Line 159: "corelation" to "correlation"
Line 203: "arrested be" to "arrested at"
etc.

Response: We thank the reviewer for positive comments as well as carefully reading. In the revised manuscript, we have corrected these points raised by the reviewer. Moreover, we have also carefully proofread the manuscript.

Reviewer #3 (Remarks to the Author):

1. The authors conclude that “PSMA8 assembles a type of testis-specific proteasome, which can actively degrade RAD51 and RPA1 at the end of meiotic prophase I during spermatogenesis. This is a crucial step for the efficient prophase I to metaphase progression in male germ cells.” (line 253). This conclusion therefore has two components, firstly, that PSMA8 is testis-specific proteasome that degrades Rad51 and Rpa1 and, secondly, that stabilised Rad51 and Rpa1 compromise efficient prophase I-to-metaphase I progression. Their data support the first component. However, as with the previous submission, it is very unconvincing that persisting stability of these two proteins compromises progression. Indeed, as they find in oocytes, Rad51 and Rpa1 ordinarily remain stable in normally progressing oocytes (Fig. 7).

Response:

- 1) We agree with the reviewer that stabilization of RAD51 and RPA1 is not the only reason that directly leads to the defects in spermatogenesis. Therefore, in the revised manuscript, we have rephrased this sentence to avoid potential misleading.
- 2) Because deletion of PSMA8 affects the stability of proteasomes in testes, we agree with the reviewer that there might be more PSMA8 substrates that remain stable in spermatocytes null for PSMA8. We have checked carefully to avoid describing that stabilized RAD51 and RPA1 compromise spermatogenesis in the revised manuscript.

- 3) Interestingly, during revision, we observed stabilization of another protein, CFP1, in PSMA8-null spermatocytes (Fig. S4e). A recent study (PMID: 30154440) has shown that CFP1 is degraded in germ cells and somatic cells when entering metaphase, and suggests physiological importance of CFP1 phosphorylation and degradation in oocyte meiosis. In this manuscript, we found that in PSMA8-deleted testes at PD42, CFP1 remained stable and exhibited a band shift, indicating that CFP1 is phosphorylated, but fails to be degraded. The stabilization of CFP1 might also contribute to the arrest of meiosis I progression in PSMA8-deleted spermatocytes.

2. The authors describe male infertility following loss of a testis-specific proteasome. However, by focusing on males, the broader interest issue of how males and females differentially regulate gametogenesis is lost.

Response: Yes, we are now focusing on the mechanism of PSMA8-associated proteasomes in regulation of meiosis progression in testes. Our results suggested that proteasomal activity is required for normal progression of spermatogenesis. Our results added novel insights into the mechanisms of proteasomal degradation during spermatogenesis.

The involvement of proteasomal degradation during spermatogenesis has been a hot topic of reproductive biology and cell biology. Many proteins such as meiotic proteins, core histones and unnecessary organelles are degraded during spermatogenesis. While the involvement of proteasomes in the degradation of core histones during late stages of spermatogenesis is well characterized, the functions of proteasomes at earlier stages in spermatocytes remain elusive. A recent study suggested that the ubiquitin-proteasome system regulates meiotic prophase I progression during spermatogenesis (Rao et. al., Science, PMID: 28059716), the physiological functions of proteasomes in meiosis I was unknown. In this study, we showed that 1) PSMA8 assembles a type of testis-specific 20S core proteasomes; 2) the activity of PSMA8-associated proteasomes is required for the degradation of meiotic proteins; 3) deletion of PSMA8 decreases the amount of proteasomes in spermatocytes from pachytene stage and causes male infertility due to defects in meiosis I progression.

These results demonstrated the importance of proteasomal activity in spermatogenesis, and add new knowledge into the functions of proteasomes and regulation of spermatogenesis. Therefore we believe that our manuscript is a strong candidate for publication in *Nature Communications*, and will draw broad interests of cell biology researchers and public readers who are interested in proteasome, human reproduction, drug design, etc.

3. Although the authors show that spermatogenesis is blocked and leads to male sterility when PSMA8 is lost, it is not at all clear why this comes about. The explanation involving Cdk1 remains very tenuous. Whilst the authors have included blots for additional Cdk1 regulators in the revised version (Fig. S4c), these data do not at all clarify how the supposed phenotype of delayed M-phase entry followed by M-phase arrest comes about or how this might be linked to deregulated proteolysis. Why is entry into M-phase delayed if Cdk1 undergoes normal activation based on pHH3 staining?

Response:

- 1) Our results suggested that deregulated proteolysis of proteins participate in meiotic prophase I (such as RAD51, RPA1, CFP1, etc.) leads to sterility in *PsmA8*-null males. This conclusion is supported by the results of accumulated ubiquitination level, decreased amount of proteasomes, and failure in degrading RAD51/RPA1/CFP1 in PSMA8-deleted spermatocytes. Thus it is clear that PSMA8 deletion causes spermatogenesis block and male sterility due to disruption of the proteasomal degradation in spermatocytes.
- 2) Phosphorylation of Histone H3 on serine 10 (pHH3), which is mediated by aurora kinases other than CDK1 kinase, is a marker of chromosome condensation. Therefore we used pHH3 as a marker of metaphase cells in both WT and PSMA8-KO testes. Based on the pHH3 staining, PSMA8-null spermatocytes showed a delay in entering metaphase (from PD21 to PD30), but were finally arrested at metaphase (PD42). The delay in entering metaphase was also supported by the persistence of diplotene cells in stage I-III tubules (Fig. 6d).
- 3) The MPF complex, which consists of CDK1 and cyclin B1, is activated at G2/M transition in mitosis and meiosis. In WT testes, the level of active form of CDK1 (pT161-CDK1) was observed in spermatocytes from late-pachytene stage and was increased in metaphase cells (Fig. S5d). Similarly, the level of pT161-CDK1 was comparably observed in PSMA8-null spermatocytes from late-pachytene stage, suggesting that meiotic prophase I progression to late-pachytene is not significantly affected by PSMA8 deletion (as well as the H1t staining in Fig. S5e). However, by Western blot, pT161-CDK1 was dramatically higher in PSMA8-null testes (Fig. S4e). This is consistent with the metaphase arrest phenotype.
- 4) We investigated more MPF regulators, but have not observed significant changes among these proteins by Western blot (cyclin B1, pY15-CDK1, EMI1, etc.). Because testes contain seminiferous tubules from stage I to XII, the Western blot results might not precisely reflect the changes in these proteins in certain types of spermatocytes. However, as long as we have tried, these antibodies did not work well for immunofluorescent staining. Therefore, it is still difficult to clarify the MPF activity in PSMA8-null spermatocytes at different stages; except for that PSMA8-null spermatocytes were finally arrested at metaphase with high T161-CDK1 phosphorylation.

4. The authors make conclusions regarding expression levels of proteins and show selected images of immunostains and Westerns. Rather, mean levels from multiple blots etc. should be quantified and depicted graphically, this would greatly simplify the interpretation. For instance, the authors claim that cyclin B1 levels don't change when PSMA8 is deleted; however, the band intensity of the +/- sample in PD21 seems more intense than for the +/- sample.

Response: To improve the interpretation of our results, we quantified the Western blot bands with ImageJ, where applicable. The mean level of each band was shown below the bands. We believe this is a standard way to present Western blot results.

5. Although the focus is now on males, the authors suggest in the Discussion that “Because spermatocytes null for PSMA8 are delayed in progressing into metaphase I, we postulate that high levels of RAD51 and RPA1 in oocytes might contribute the dictyate arrest (or GV arrest) during the processes of follicle formation.” This is unfounded as spermatogenesis in testes is in no way comparable to the G2-arrest in oocytes that involves a unique inhibitory follicular environment for each individual oocyte combined with oocyte-specific APC-Cdh1-mediated cyclin B proteolysis.

Response: We thank the reviewer for the constructive comment, and have removed the related description in the revised manuscript.

Overall, the authors show that knocking out PSMA8 results in a late spermatogenic block and male infertility. This is based on good-quality immunofluorescence but at this stage does not provide any clear insight into why proteasomal disruption in this particular model leads to a meiotic arrest in M-phase.

Response: Since DSB repair and meiotic recombination were not affected by PSMA8-deletion and PSMA8-null spermatocytes progressed normally to late-pachytene stage (as shown by H1t staining in the revised Fig. S6a), our results strongly suggested that proteasomal degradation is crucial for prophase I to anaphase I progression during spermatogenesis. Because proteasomes degrades many ubiquitinated and un-ubiquitinated proteins during spermatogenesis, we believe that the M-phase arrest phenotype is a complicated readout of deregulated proteasomal degradation (RAD51, RPA1 and CFP1) in PSMA8-null spermatocytes.

Reviewers' comments:

Reviewer #1 (Remarks to the Author):

The authors have addressed most of the concerns adequately, and therefore the manuscript is now acceptable for publication.

Reviewer #3 (Remarks to the Author):

Here the authors provide data supporting that PMSA8 largely replaces PMSA7 as a core 20S proteasomal component during spermatogenesis. They deleted PMSA8 in males and found that spermatogenesis arrests during M-phase of meiosis I (MI). RAD51, RPA1 and in the revised version, CFP1, levels appear to be higher after PMSA8 deletion. Although entry into M-phase appears delayed, it is not prevented, despite stabilisation of RAD51 and RPA1. Evidence of M-phase arrest is that spermatids are absent from seminiferous tubules and markers of activated Cdk1 (pT161-Cdk1) remain persistently high.

Comments:

1. The authors provide evidence that the levels of some proteins are higher after PMSA8-deletion. Throughout the paper the authors then assume that this is because these proteins are not being properly degraded (e.g. lines 170-182). To prove that higher levels are specifically due to impaired proteolysis (and not, for instance, altered translation), requires direct proof that these proteins are actually degraded in the first place e.g. by using MG132. It seems that this has been done only once, for RAD51 (Supp Fig. 4b); a single blot is presented and band intensity quantified and the difference does not look convincing. The reproducibility of this change needs to be proven with a mean intensity and statistical comparison from at least 3 blots to show that levels are significantly different following MG132 treatment.
2. Data support that some proteins involved in prophase I are higher following PMSA8-deletion (e.g. RAD51 and RPA). However, the block to spermatogenesis is not in prophase I (albeit there appears to be some delay in M-phase entry), it is in M-phase of MI. It therefore seems that although these prophase I proteins may be stabilised, it is of no real physiological consequence to prophase I (see Fig 5 and lines 207-211) in further support of which, prophase I in oocytes progress normally although both proteins remain stable.
3. The important question is why does M-phase become arrested following PMSA8-deletion and how is this linked to PMSA8-dependent proteolysis? The authors show higher levels of CFP1 and speculate that this could be responsible for M-phase arrest based on findings in oocytes. But since oogenesis and spermatogenesis exhibit very stark differences in underlying mechanisms (which is one of the points of this paper), this is a very tenuous line of evidence.
4. Throughout the paper the authors refer to "metaphase" when it should be M-phase. Metaphase is a specific point within M-phase when chromosomes have become aligned at the equator of the bipolar spindle. Since spindles and chromosomes have not been studied in any detail, it is not known what stage of M-phase arrest occurs.

Overall, the authors undertake extensive and very high-quality immunofluorescence staining of seminiferous tubules. They show for the first time that PMSA8 is required for progression through M-phase during spermatogenesis but there isn't a coherent explanation for WHY this occurs. The strongest evidence that PMSA8-deletion affects protein levels is for proteins involved in prophase I, but this seems of limited physiological consequence. The evidence that increased protein levels are specifically due to impaired proteolysis requires more extensive corroboration – portraying a single blot is not sufficient to prove reproducibility and significance of difference. The most interesting defect following PMSA8-deletion is an M-phase arrest for which, no robust explanation is provided. The most glaring gap in this paper therefore pertains to a lack of explanation for why this happens and what proteolytic event is required to be performed by PMSA8 to ensure progression beyond M-phase.

Reviewer #3 (Remarks to the Author):

1. The authors provide evidence that the levels of some proteins are higher after PMSA8-deletion. Throughout the paper the authors then assume that this is because these proteins are not being properly degraded (e.g. lines 170-182). To prove that higher levels are specifically due to impaired proteolysis (and not, for instance, altered translation), requires direct proof that these proteins are actually degraded in the first place e.g. by using MG132. It seems that this has been done only once, for RAD51 (Supp Fig. 4b); a single blot is presented and band intensity quantified and the difference does not look convincing. The reproducibility of this change needs to be proven with a mean intensity and statistical comparison from at least 3 blots to show that levels are significantly different following MG132 treatment.

Response: In the revised manuscript, we have provided the quantification of RAD51 from 4 different Western blots. For each blot, the highest RAD51 level was set as "1" and the rest of the samples were normalized to this sample. The average expression level was shown in Fig. S4b. From the Western blot of RAD51 and its quantification, we show that RAD51 in WT testes was degraded by proteasomal degradation.

2. Data support that some proteins involved in prophase I are higher following PMSA8-deletion (e.g. RAD51 and RPA). However, the block to spermatogenesis is not in prophase I (albeit there appears to be some delay in M-phase entry), it is in M-phase of MI. It therefore seems that although these prophase I proteins may be stabilised, it is of no real physiological consequence to prophase I (see Fig 5 and lines 207-211) in further support of which, prophase I in oocytes progress normally although both proteins remain stable.

Response:

(1) Many mouse models showed M-phase arrest because of the defects in prophase I (*Tex11^{-/-}*, *Hfm1^{-/-}*, *Rnf212^{-/-}*, *Hei10^{-/-}*, etc.). Protein products of these genes are required for meiotic recombination and prophase I progression. Deletion of these genes in mouse results in insufficient formation of crossovers at prophase I during spermatogenesis. However, the resulting spermatocytes are arrested at M-phase of MI, instead of prophase I arrest, suggesting that certain defects in prophase I could cause M-phase arrest.

(2) In our PSMA8 case, although crossover formation was not affected, some prophase I proteins remained stable in PSMA8-deleted spermatocytes due to impaired proteolysis. In line with this observation, PSMA8-deleted spermatocytes progressed slower into M-phase (Fig. 6c-d) and were arrested at M-phase of MI. Therefore we proposed that impaired degradation of these prophase I proteins might be, at least, part of the reason that contributes to the M-phase phenotypes.

(3) Because oocyte prophase I is not synchronized in developing ovaries, it is difficult to stage the progression of prophase I in ovaries, especially when prophase I progression is not

significantly affected. As long as female fertility is not changed by PSMA8 deletion, we did not carefully characterize the prophase I progression in meiotic oocytes.

3. The important question is why does M-phase become arrested following PSMA8-deletion and how is this linked to PSMA8-dependent proteolysis? The authors show higher levels of CFP1 and speculate that this could be responsible for M-phase arrest based on findings in oocytes. But since oogenesis and spermatogenesis exhibit very stark differences in underlying mechanisms (which is one of the points of this paper), this is a very tenuous line of evidence.

Response: In this manuscript, we proposed that defects in proteasomal degradation of prophase I proteins leads to the phenotypes in meiosis I. We provided evidences of several proteins, which remain stable in PSMA8-deleted testes. According to the M-phase phenotypes observed in above-mentioned prophase I-deficient mouse models and the hyperactivation of pT161 CDK1 in PSMA8-deleted spermatocytes, we proposed that ectopic accumulation of prophase I proteins contributes to the M-phase phenotypes in spermatogenesis.

Because testis is a mixture of various somatic cells, spermatocytes and spermatids, we are not able to profile the proteome-wide alternation of protein levels. On the other hand, we agree with the reviewer that the rationale for CFP1 in spermatogenesis is weak and therefore removed the results related to CFP1 in the revised manuscript.

4. Throughout the paper the authors refer to “metaphase” when it should be M-phase. Metaphase is a specific point within M-phase when chromosomes have become aligned at the equator of the bipolar spindle. Since spindles and chromosomes have not been studied in any detail, it is not known what stage of M-phase arrest occurs.

Response: We thank the reviewer for carefully reading. And yes, we agree that it should be M-phase instead of metaphase, given that chromosomes are not always aligned well in those M-phase arrested cells. We have changed “metaphase” to “M-phase” accordingly in the revised manuscript.

REVIEWERS' COMMENTS:

Reviewer #3 (Remarks to the Author):

The authors have made the suggested changes to improve the manuscript. I'm assuming that there was a typo in the response letter since Western quantifications are shown in Fig. S5b rather than in S4b.

Reviewer #3 (Remarks to the Author):

The authors have made the suggested changes to improve the manuscript. I'm assuming that there was a typo in the response letter since Western quantifications are shown in Fig. S5b rather than in S4b.

Response: We thank the reviewer for positive comments and carefully reading. And yes, it should be Fig. S5b in the revised manuscript, since we have added a Fig. S2.